# Differential LysoTracker Uptake Defines Two Populations of Distal Epithelial Cells in Idiopathic Pulmonary Fibrosis

**DOI:** 10.3390/cells11020235

**Published:** 2022-01-11

**Authors:** Roxana Maria Wasnick, Irina Shalashova, Jochen Wilhelm, Ali Khadim, Nicolai Schmidt, Holger Hackstein, Andreas Hecker, Konrad Hoetzenecker, Werner Seeger, Saverio Bellusci, Elie El Agha, Clemens Ruppert, Andreas Guenther

**Affiliations:** 1Universities of Giessen and Marburg Lung Center (UGMLC), The German Center for Lung Research (DZL), 35392 Giessen, Germany; irn.bry@yandex.ru (I.S.); jochen.wilhelm@innere.med.uni-giessen.de (J.W.); Ali.Khadim@innere.med.uni-giessen.de (A.K.); nschmidt97@hotmail.de (N.S.); Werner.Seeger@innere.med.uni-giessen.de (W.S.); saverio.bellusci@innere.med.uni-giessen.de (S.B.); Elie.El-Agha@innere.med.uni-giessen.de (E.E.A.); Clemens.Ruppert@innere.med.uni-giessen.de (C.R.); Andreas.Guenther@innere.med.uni-giessen.de (A.G.); 2Excellence Cluster Cardiopulmonary Institute (CPI), 35392 Giessen, Germany; 3Max-Planck-Institute for Heart and Lung Research, 61231 Bad Nauheim, Germany; 4Institute for Lung Health (ILH), 35392 Giessen, Germany; 5Department of Clinical Immunology and Transfusion Medicine, 35392 Giessen, Germany; Holger.Hackstein@uk-erlangen.de; 6Department of General and Thoracic Surgery, University Hospital Giessen, 35392 Giessen, Germany; Andreas.Hecker@chiru.med.uni-giessen.de; 7Department of Thoracic Surgery, Medical University of Vienna, 1090 Vienna, Austria; konrad.hoetzenecker@meduniwien.ac.at; 8European IPF Registry/UGLMC Giessen Biobank, 35392 Giessen, Germany; 9Lung Clinic Waldhof-Elgershausen, 35753 Greifenstein, Germany

**Keywords:** IPF, alveolar epithelial cells, intermediate epithelial cells, transitional states, LysoTracker, flow cytometry, lung transcriptomic profile, CK5, NGFR, CD24

## Abstract

Idiopathic pulmonary fibrosis (IPF) is a progressive and fatal degenerative lung disease of unknown etiology. Although in its final stages it implicates, in a reactive manner, all lung cell types, the initial damage involves the alveolar epithelial compartment, in particular the alveolar epithelial type 2 cells (AEC2s). AEC2s serve dual progenitor and surfactant secreting functions, both of which are deeply impacted in IPF. Thus, we hypothesize that the size of the surfactant processing compartment, as measured by LysoTracker incorporation, allows the identification of different epithelial states in the IPF lung. Flow cytometry analysis of epithelial LysoTracker incorporation delineates two populations (Lyso^high^ and Lyso^low^) of AEC2s that behave in a compensatory manner during bleomycin injury and in the donor/IPF lung. Employing flow cytometry and transcriptomic analysis of cells isolated from donor and IPF lungs, we demonstrate that the Lyso^high^ population expresses all classical AEC2 markers and is drastically diminished in IPF. The Lyso^low^ population, which is increased in proportion in IPF, co-expressed AEC2 and basal cell markers, resembling the phenotype of the previously identified intermediate AEC2 population in the IPF lung. In that regard, we provide an in-depth flow-cytometry characterization of LysoTracker uptake, HTII-280, proSP-C, mature SP-B, NGFR, KRT5, and CD24 expression in human lung epithelial cells. Combining functional analysis with extracellular and intracellular marker expression and transcriptomic analysis, we advance the current understanding of epithelial cell behavior and fate in lung fibrosis.

## 1. Introduction

The human lung is a highly complex organ designed specifically for gas exchange. In idiopathic pulmonary fibrosis (IPF), chronic epithelial injury leads to excessive deposition of rigid extra-cellular matrix and a progressive decrease in lung compliance and gas-exchange surface, causing inevitable and fatal lung failure within 2–5 years after diagnosis [1,2,3]. Although new therapies significantly increased the duration and quality of life of IPF patients, a therapeutic regimen that can arrest or, even better, reverse disease progression remains to be discovered [4]. Partly responsible for this situation is our limited understanding of the cellular states and processes that each of the more than 40 cell types in the lung undergoes, in an active (causative) or reactive manner in homeostatic and injury contexts [5,6]. A number of recent studies clearly identified the chronic injury of the alveolar type 2 epithelial cells (AEC2s) as the initial site of injury in the IPF lung [7,8,9]. AEC2s are facultative progenitors in the distal lung, which, in a differentiated state, serve the vital function of surfactant production and secretion, but can also act as progenitors for other AEC2s and AEC1s in homeostatic and injury-repair situations [10,11].

Pulmonary surfactant is a phospholipoprotein mixture secreted exclusively by AEC2s which reduces the alveolar surface tension necessary for alveoli reopening during the respiratory cycle. The protein component is represented by the surfactant proteins (SP) A, B, C and D, with two of them, SP-B and SP-C, holding tension-active properties. Following the processing from the pro- forms (proSP-B and proSP-C) to the mature forms (mSP-B and mSP-C), they are secreted in their mature forms specifically by AEC2s [12,13,14]. The processing and assembly of pulmonary surfactant proteins take place in the lamellar bodies of AEC2s, specialized organelles characterized by very low pH [15].

In IPF, repeated alveolar injury results in the recruitment of the AEC2 progenitors necessary for the repair process [11,16,17,18]. In this process, the differentiated function of bona fide AEC2s, defined as AEC2s which synthesize, process, and secrete alveolar surfactant, is impaired and leads to increased alveolar surface tension and increased alveolar collapse, which propagates the injury even further, thus creating a self-propagating cycle of injury and repair [19,20,21,22]. The acute AEC1/2 injury creates a microenvironment where other reactive cell types, such as alveolar macrophages and fibroblasts, are quickly activated and recruited to cover the basement membrane and prevent fluid leakage into the airspace [23]. However, as the AEC2 progenitor pool is exhausted by injury or by extensive proliferation, the long-term repair after or during the chronic and repeated injury relies on the recruitment of other local epithelial progenitors, several of which have already been identified in the mouse lung [17,24,25,26]. In humans, the profound histological changes found in the distal IPF lung are consistent with the expansion of a cytokeratin 5 (CK) progenitor, but its origin and differentiating trajectory remain to be determined [27]. Recent landmark papers described the transcriptomic signatures of disease-free (donor) and IPF epithelial cells at single-cell level, leading to the identification of transcriptomic signatures for many known epithelial cell types in the lung and the identification of novel ones (ionocytes and CK17+/CK5- aberrant AEC2s) [16,28,29,30,31,32]. However, it is unclear how these transcriptomic signatures translate into the stable or transitional cellular states and processes responsible for the disease phenotype [33]. The repair process in IPF is ultimately ineffective, underlying the disease progression that leads to organ failure. Thus, the ability to correlate scNGS data with protein expression and functional behavior would greatly increase our understanding of these epithelial fates and states and turn this into a therapeutically actionable process for the benefit of IPF patients.

Here, we ask if the aberrant or intermediate transcriptional programs recently identified in IPF [29,31] result in functional transitional states that can be identified by the low/intermediate ability to process and secrete surfactant proteins (SP). To that end, we analyze the size of the surfactant processing compartment in dissociated human donor and IPF lung epithelial cells, thus defining two functional alveolar epithelial states present in both donor and IPF lung. Based on known intracellular cell surface proteins and LysoTracker incorporation, coupled with transcriptomic analysis, we show that the LysoTracker^high^ population consists of bona fide AEC2s and is drastically diminished in IPF. A second population of LysoTracker^low^ cells, which uniformly expresses and processes surfactant proteins but bears the transcriptional footprint of a CK5-derived (basal) population, is increased in IPF.

## 2. Materials and Methods

### 2.1. Animal Studies

Animal studies were performed in accordance with the Helsinki convention for the use and care of animals and were approved by the local authorities at Regierungspräsidium Giessen V54-19 c 2015 (1) GI 20/10 Nr. 109/2011 (Bleomycin) or V54-19 c 20 15 h 02 GI 20/10 Nr. A53/2012 (untreated controls).

### 2.2. Patient Material

The study protocol was approved by the Ethics Committee of the Justus-Liebig-University School of Medicine (No. 31/93, 29/01, and No. 111/08: European IPF Registry), and informed consent was obtained in written form from each subject. Explanted lungs (*n* = 31 for sporadic IPF, IPF_LTX_; *n* = 6 for COPD) or non-utilized donor lungs or lobes fulfilling transplantation criteria (*n* = 27; human donors) were obtained from the Dept. of Thoracic Surgery in Giessen, Germany and Vienna, Austria and provided by the UGMLC Giessen Biobank, a member of the DZL platform Biobanking. All IPF diagnoses were made according to the American Thoracic Society (ATS)/European Respiratory Society (ERS) consensus criteria [34], and a usual interstitial pneumonia (UIP) pattern was proven in all IPF patients.

### 2.3. Bleomycin Model of Lung Fibrosis

C57BL/6N mice (Charles River Laboratories, Sulzfeld, Germany) between 10 and 16 weeks old were used. Mice were intubated and bleomycin (Hexal, 2.5U/Kg body weight in 0.9% saline) was aerosolized using a microsprayer (Penncentury). At each time point, saline-treated and/or untreated mice were used as controls. Mice were weighed every day and sacrificed 3, 7, 14, 21 and 28 days later for cell dissociation and flow cytometry analysis.

### 2.4. Lung Tissue Dissociation

For both mouse and human lung, standard dispase-based dissociation protocols were used, as previously described [22,35,36] and detailed in the Appendix A.

### 2.5. Flow Cytometry Analysis and Cell Sorting

Standard [37], previously published methods [22,35,36] were used for sample preparation and intracellular and extracellular staining in preparation for flow cytometry and fluorescence-activated cell sorting. Detailed methods and reagents, including all antibodies, are described in the Appendix A. Single color controls were used to compensate for spectral overlap. Fluorescence-minus-one (FMO) controls were used whenever possible for positive/negative population gating. In the case of indirect intracellular staining, no primary control samples, consisting of the FMO control in the particular channel to which only the secondary antibody was added, were used for data interpretation and quantification. Data were acquired on a BD FACSCanto II (BD Biosciences) using BD FACSDiva software (BD Biosciences). Data were further analyzed using FlowJo vX software (FlowJo, LLC).

### 2.6. Immunofluorescence Analysis

The staining procedures were based on standard, previously published techniques and the reagents are listed in the Appendix A. However, given that both the mature SP-B and proSP-B antibodies were raised in the same species (rabbit), the standard protocol was modified, as follows. Following standard deparaffinization and blocking (see Appendix A), slides were incubated with a mature SP-B antibody at a very low concentration (1:2000, 10 times lower than for traditional mature SP-B staining) and the fluorescent signal was amplified using the Alexa Fluor™ 555 Tyramide SuperBoost™ Kit, goat anti-rabbit IgG (Thermo Scientific, Waltham, MA, USA). This resulted in the covalent attachment of Alexa Fluor 555 Tyramide at the base of the antigen, which allowed the consequent stripping of the rabbit anti-mature SP-B antibody using the standard citrate-based antigen retrieval solution, as described in the Appendix A. Samples were re-blocked with 5% BSA in PBS solution and incubated with the rabbit anti-proSP-B antibody, followed by an Alexa Fluor 488-labeled donkey anti-rabbit secondary antibody. Appropriate controls demostrating the lack of cross-reactivity were used to ascertain the specificity of the two signals (see Appendix A). The stained amsples were imaged on a wide-field fluorescence microscope (Axio Observer.Z1 fluorescence microscope, Carl Zeiss MicroImaging, Jena, Germany) and a confocal microscope (TCS SP5, Leica Microsystems) and the images were processed and quantified using the Fiji package of ImageJ image analysis software ((https://imagej.net, version 20.0-rc-65/1.51w, accessed on 31 January 2018).

### 2.7. Microarray Analysis

Purified total RNA was amplified using the Ovation PicoSL WTA System V2 kit (NuGEN Technologies, Bemmel, The Netherlands). For each sample, 2 µg of amplified cDNA was Cy3-labeled using the SureTag DNA labeling kit (Agilent, Waldbronn, Germany). Hybridization to 8 × 60K 60mer oligonucleotide spotted microarray slides (Human Whole Genome, SurePrint G3 Human GE v3 8 × 60K Microarray; Agilent Technologies, design ID 072363) and the subsequent washing and drying of the slides was performed following the Agilent hybridization protocol in Agilent hybridization chambers with the following modifications: 2 µg of the labeled cDNA were hybridized for 22 h at 65 °C. The cDNA was not fragmented before hybridization. The dried slides were scanned at a 2 µm/pixel resolution using the InnoScan is900 (Innopsys, Carbonne, France). Image analysis was performed with Mapix 8.2.5 software, and calculated values for all spots were saved as GenePix result files. Stored data were evaluated using the R software (www.r-project.org, version R3.6.3 GUI1.70 ElCapitan, accessed on 29 February 2020) and the limma package [38] from BioConductor [39]. Log_2_ mean spot signals were taken for further analysis. Data were background corrected using the NormExp procedure on the negative control spots and were quantile-normalized [38,40] before averaging. Log_2_ signals of replicate spots were averaged, and, from several different probes addressing the same gene, only the probe with the highest average signal was used. Genes were ranked for differential expression using a moderated t-statistic [38]. Pathway analyses were performed using gene set tests on the ranks of the t-values [38,41]. Pathways were taken from the KEGG database (http://www.genome.jp/kegg/pathway.html, accessed on 11 October 2021).

Heatmaps are generated from the normalized log_2_ spot intensities (*I*) and show the gene-wise z-values (where $z_{j}=\left(I_{j}-mean\left(I\right)\right)/SD\left(I\right)$ for j=1…n).

### 2.8. Data Analysis

#### 2.8.1. Flow Cytometry Data Analysis

The frequency of parent and mean fluorescence intensity (MFI) data were exported from FlowJo v.10 and analyzed using MicrosoftExcel, R software (www.r-project.org, version R3.6.3 GUI1.70 ElCapitan, accessed on 29 February 2020) or GraphPad Prism (GraphPad Software). For the bleomycin experiments, the percentages of Lyso^high^ and Lyso^low^ cell populations in control vs. treated samples were first log-transformed and Student’s T-test was used to determine the statistical significance of their differences at each time point. To evaluate the dynamic of the Lyso^high^ and Lyso^low^ populations over time (Figure 1C), we related the proportion of each Lyso^high^ or Lyso^low^ population in each group (bleomycin or saline) to the total LysoTracker-incorporating population in the control animals at each time point, as follows: each point on the line represents the % change log10 of the respective Lyso^high^ or Lyso^low^ at DayX, calculated as log10(Lyso^high or low^) DayX_bleo or control_/log10(Lyso^high^DayX + Lyso^low^DayX)_control_. To determine the relative difference between the Lyso^high^ and Lyso^low^ populations in each epithelial group at each time point (Figure 1D), we analyzed the log odds ratio of these two populations using the R statistical analysis software. A log odds ratio of 0 means that the two populations are similar, and, therefore, the probability that an epithelial cell is a Lyso^high^ cell is equal to that of it being a Lyso^low^ cell. Positive log odds ratios indicate that the two populations are different, and, therefore, the probability of a cell belonging to one population is larger than the probability of it belonging to the other one. Flow cytometry data collected from human samples were analyzed in a similar manner. Student’s t-tests or two-way ANOVAs were used, as appropriate (stated in the figure legend), to test the null hypotheses that the log-transformed MFI of the percentage of parent values were different in each comparison.

#### 2.8.2. Immunoflurescence Quantification and Analysis

Fluorescence intensity was analyzed using the Fiji/Image J (https://imagej.net, version 20.0-rc-65/1.51w, accessed on 31 January 2018) image analysis software and fluorescence intensity data was exported and statistically analyzed and plotted using Microsoft Excel.

## 3. Results

### 3.1. LysoTracker Incorporation Delineates Two Populations of Epithelial Cells in Bleomycin-Induced Injury

To determine the dynamic behavior of AEC2s during bleomycin injury, C57BL6 mice were treated with bleomycin (2.5U/kg) and analyzed 3, 7, 14, 21, and 28 days post-administration. Saline or untreated mice (generally termed controls) were used as controls at each time point. Mice were sacrificed and their lungs were dissociated into a single-cell suspension whose cellular composition was analyzed by flow cytometry at each time point. To identify AEC2s, dead cells (PI^+^ or DAPI^+^) and AEC1 cells (podoplanin-PDPN^+^) were first excluded, and the epithelial compartment was further identified by EpCAM expression within the CD45^−^ (non-hematopoietic) and CD31^−^ (non-endothelial) population (full gating path is in Appendix A). The proportion of the DAPI^−^ CD45^−^ CD31^−^ PDPN^−^ EpCAM^+^ population (the epithelial cell compartment from here on) was slightly decreased in bleomycin-treated mice starting on day 14 and reached statistical significance on day 28 (Appendix A). To identify AEC2s within the epithelial compartment, we took advantage of their specific ability to uptake LysoTracker dyes [12,42]. The dynamic of LysoTracker uptake was analyzed during the bleomycin recovery time course, revealing three distinct populations: a LysoTracker^neg^ (Lyso^neg^), a LysoTracker^low^ (Lyso^low^) and a LysoTracker^high^ (Lyso^high^) population (Figure 1A). At all time-points analyzed, the Lyso^high^ population was decreased in number, with the greatest decrease registered at days 7 and 14, when AEC2 injury was maximal [43,44]. This was paralleled by a proportional increase in the Lyso^low^ population that reached a maximum increase at the same time points (Figure 1B). The time-course analysis of the population dynamic showed that Lyso^high^ and Lyso^low^ populations behaved complementary to each other, with a maximum relative change at day 7 and a partial recovery by days 21 and 28. The paired analysis of the log odds ratio of the Lyso^high^ and Lyso^low^ populations (log Lyso^high^/Lyso^low^) further supported our conclusion that the difference between the two populations was maximal in control samples and early time points but decreased significantly at days 7 and 14 (log value of zero) (Figure 1D). Of note, LysoTracker uptake was not completely recovered at day 28, suggesting long-lasting alterations in cellular phenotype (Figure 1B–D).

### 3.2. LysoTracker Uptake in Human Lung Epithelium

Next, we asked if the behavior of these populations is similar in the distal human donor and IPF lung. To that end, subpleural tissue from six donor and six end-stage IPF explanted lungs were dissociated into single-cell suspensions and analyzed by flow cytometry. The gating strategy was similar to that of the mouse lung, where the epithelium was identified as live CD45^−^ CD31^−^ EpCAM^+^ cells (Appendix A). Fluorescence-minus-one (FMO) samples were used for appropriate gating (Appendix A). There was no statistically different proportion of epithelial cells between the groups (Appendix A). Similar to the mouse data, the proportion of IPF Lyso^high^ cells was dramatically decreased compared to donors, from an average of 50.3% in donors to 10.1% in IPF. The Lyso^low^ population behaved in a complementary fashion, increasing from an average of 15.2% in donors to 37.1% in IPF patients (Figure 2A,B). Individual panels from each patient are shown in Appendix A.

To further understand the identity of the Lyso^high^- and Lyso^low^-incorporating cells, we analyzed the expression of HTII-280, a well-known AEC2 marker, as a function of LysoTracker incorporation [45]. In addition to donor (*n* = 6) and IPF (*n* = 7) samples, COPD samples (*n* = 7) were added as non-IPF related controls. Quadrant gating of LysoTracker versus HTII-280 expression in the epithelial cell population, as in Figure 2A, led to the identification of four populations: Q1 (Lyso^pos^/HTII-280^neg^), Q2 (Lyso^pos^/HTII-280^pos^), Q3 (Lyso^neg^/HTII-280^pos^), and Q4 (Lyso^neg^/HTII-280^neg^) (Figure 2C). In donor and COPD samples, the largest proportion of epithelial cells comprised bona fide AEC2s (Q2: 79.64% Donor and 67.39% COPD), which were Lyso^pos^/HTII-280^pos^. In contrast, in the IPF samples, the proportion of Q2 cells was markedly reduced to an average of 14.33%, consistent with the well-established chronic injury of AEC2s characteristic of IPF. This decrease in Q2 was paralleled by a marked increase in Q1, which represents the Lyso^pos^/HTII-280^neg^ cells, from 5.12% in donor samples and 9.43% in COPD samples to 47.73% in IPF (Figure 2D,E). Populations Q3 and Q4 were not significantly altered in all comparisons, with the exception of a slight but statistically increase in Q4 in the comparison of COPD and IFP samples (Appendix A). The analysis was very consistent from patient to patient, with some variability noted in the Q4 (Lyso^neg^/HTII-280^pos^) population, as shown in Appendix A.

At a first glance, the gating strategy suggests that the Q2 population consists mostly of Lyso^high^ cells, while the Lyso^low^ cells belong to Q1. Thus, we compared the mean fluorescence intensity (MFI) of the LysoTracker-incorporating populations Q1 and Q2, which showed a constant and statistically significant increase in LysoTracker incorporation in Q2 compared to Q1, demonstrating that Q2 comprises mostly Lyso^high^ cells and Q1 comprises mostly Lyso^low^ cells. This difference was maintained in all three groups, regardless of their disease status (Figure 2F), suggesting that these two parameters define two distinct cellular states and, in this regard, functionally homogeneous populations (Figure 2G,H). Taken together, our data suggest the existence of two distinct epithelial populations with distinct LysoTracker uptake characteristics that vary in an inversely correlated manner, suggestive of compensatory behavior in IPF patients compared to donors. Moreover, the Lyso^high^ population is marked by the well-established HTII-280 antibody, confirming its bona fide AEC2 identity.

### 3.3. Surfactant Protein Expression Defines Two Populations of AEC2s in Donor and IPF Lung

Surfactant protein production, processing and secretion is the most defining characteristic of AEC2s. Thus, we asked what the pattern of proSPC expression is in relation to HTII-280. To that end, following the usual cell surface staining (CD45, CD31, EpCAM, and HTII-280), the same six donor and six IPF single-cell preparations used in the previous analysis were fixed, permeabilized, and stained intracellularly with a proSP-C specific antibody. Because the LysoTracker signal is lost during the fixation process, we relied on HTII-280 reactivity for the identification of the bona fide AEC2s (DAPI^neg^ CD45^neg^ CD31_neg_ EpCAM^pos^ HTII-280^pos^). Analysis of HTII-280 vs. proSPC expression in these samples resulted in four populations: Q1 (proSP-C^pos^/HTII-280^neg^), Q2 (proSP-C^pos^/HTII-280^pos^), Q3 (proSP-C^neg^/HTII-280^pos^) and Q4 (proSP-C^neg^/HTII-280^neg^) (Figure 3A and Appendix A). Similar to the Lyso/HTII-280 analysis (Figure 2), donor bona fide AEC2s (Q2, proSP-C^pos^/HTII-280^pos^) were the highest represented population (Q2 = 66.7%) and their proportion was markedly decreased to 6.75% in IPF samples (Figure 3B). Moreover, a population that was HTII-280^neg^ but expressed lower levels of proSP-C than Q2 was present in both patient groups, and it was markedly increased in IPF (donor Q1 = 17.31% vs. IPF Q1 = 33.84%, Figure 3B,C). Confirming previously known data, analysis of the amount of proSP-C expressed, as measured by the proSP-C MFI of each population, showed that in IPF the bona fide AEC2s (Q2) expressed significantly less proSP-C compared to donors. However, the Q1 (proSP-C^pos^/HTII-280^neg^) population, which was increased proportionally in IPF patients, did not differ in the amount of proSP-C expressed (Figure 3B–D). This suggests that, while in IPF the number and SP-producing function of AEC2s is decreased, the potentially compensatory proSP-C^low^ HTII-280^neg^ (Q1) population expresses lower levels of proSP-C. Additionally, there was a statistically significant increase in the proSP-C^neg^ HTII-280^neg^ (Q4) population, which suggested the increased presence of non-AEC2 cells in the distal IPF lung (Figure 3D). The Q3 population, representing proSP-C^neg^ HTII-280^pos^ cells was negligible and did not vary significantly with the disease state (Appendix A).

Although characteristic for the alveolar epithelium, expression of proSP-C and proSP-B has been previously noted in the non-alveolar compartment of the human lung. However, only AEC2s have the unique ability to process and secrete the mature forms (mSP-C and mSP-B). Thus, we analyzed the expression of mature SP-B (mSP-B) by intracellular staining of the same donor (*n* = 6) and IPF (*n* = 6) samples as in the previous analyses in conjunction with the usual cell surface markers. Similar to the proSP-C data, mSP-B was expressed in the majority of the bona fide AEC2s (Q2: HTII-280^pos^ mSP-B^pos^) in both donor and IPF, and their proportion was drastically reduced in IPF (Figure 3E,F and Appendix A). However, the IPF Q1 (HTII-280^neg^ mSP-B^pos^) population expressed higher levels of mSP-B than donor Q1, suggesting an upregulation of the surfactant processing ability in this population in disease conditions. Of note, the expression level of mSP-B in Q1 of IPF remained below that of Q2 (HTII-280^pos^ mSP-B^pos^), suggesting a distinct functional state of this population (Figure 3E–H). Similar to the proSP-C data, there was no significant change in the proportion of the Q3 (HTII-280^pos^ mSP-B^neg^) population but there was a significant increase in Q4 (HTII-280^neg^ mSP-B^neg^) cells.

Throughout our analysis, we noticed very consistent similarities among the LysoTracker, proSP-C and mSP-B expression pattern in relation to HTII-280: the Q1 and Q2 populations behaved similarly in all samples in each analysis. The co-staining of LysoTracker with intracellular markers is technically not feasible because of the loss of LysoTracker fluorescence during the fixation/permeabilization process necessary for intracellular staining. However, comparative and concomitant analysis of the Q1–Q4 profile with the three markers in the same donor (*n* = 6) and IPF (*n* = 6) patient samples showed that the proportion of cells belonging to Q1–4 in each population was very similar in the three parallel analyses (Figure 4A). This suggested, in a correlative manner, that the Q1 population represents a Lyso^low^, proSP-C^low^, mSP-B^low^ population of AEC2-like cells while the Q2 population represents the Lyso^high^, proSP-C^high^, mSP-C^high^ population of bona fide AEC2s. To confirm the existence of mSP-B expressing cells outside of the LysoTracker-incorporating compartment, donor and IPF peripheral lung tissue sections were co-stained for mSP-B and ABCA3, a protein specifically expressed in the lamellar bodies of mature AEC2s. Indeed, in the donor lung, mSP-B was present in almost all ABCA3-expressing AEC2s, while in IPF extensive epithelial areas (identified morphologically) were characterized by mSP-B expression in the absence of ABCA3 (Figure 4B). The expression of LysoTracker and proSP-C was very consistent within each patient group (Do vs. IPF and Appendix A). However, the intensity of the mSP-B staining was highly variable within the same patient group, suggesting the existence of a variable mSP-B processing capacity (Figure 4A and Appendix A). To confirm that the variable processing ability of AEC2 cells was not an artefact of the cell isolation procedure, donor (*n* = 4) and IPF (*n* = 4) paraffin-embedded tissue sections were co-stained for proSP-B and mSP-B, and the fluorescence intensity of each was quantified. A linear regression analysis showed that the processing ability of each sample, represented by the regression’s slope, was variable within each group, but an overall flattening of the slope was noted between IPF and donor samples. Moreover, the two values yielded were positively correlated in most donor samples (positive R^2^ values), but this correlation was lost in three out of the four IPF samples (R^2^ = 0, Figure 4C,D). When analyzing the spatial distribution of the two signals, we noticed that in donors they were tightly co-expressed (Figure 4D upper panels), but in IPF there was a heterogeneous distribution of the areas of mature and proSP-B co-localization (particularly in non-affected areas, Figure 4D panels in rows 2 and 4) and areas where the mSP-B was low or absent in cells that expressed the pro forms (Figure 4D—panels in row 3). Together, these data show that while it is variable in donor samples, mSP-B processing ability is decreased in IPF.

### 3.4. Transcriptional Characterization of the IPF Lyso^low^ Population

Given the recent single-cell NGS data that identified the existence of transitional AEC2 states with distinct transcriptomic signatures in normal and IPF lungs [28,29,30,31,46], we asked if the Lyso^low^ population in IPF, which has an intermediate expression profile in terms of LysoTracker and surfactant protein expression, resembled any of the previously mentioned intermediate populations. Thus, we used microarray analysis to compare the transcriptomic profile of eight FACS-sorted donor Lyso^pos^ AEC2s, composed, in the majority, of Lyso^high^ cells (see Figure 2A), and Lyso^pos^ cells from six IPF lungs, consisting, in the majority, of Lyso^low^ cells (Figure 2A and Figure 5A). Principal component analysis of the data showed the lack of variance between the two groups (Appendix A), suggesting great similarities between the two populations. However, in this analysis, 612 genes were upregulated (LFC > 2) and 1382 genes were downregulated (LFC < −2) in IPF Lyso^pos^ compared to donor Lyso^pos^ AEC2s. Interestingly, the first 50 upregulated genes in the Lyso^pos^ population of IPF patients included several genes known to be upregulated in IPF while several surfactant-related genes were noted in the 50 most downregulated genes (Figure 5B and Appendix A). Validating our data, KEGG analysis identified metabolic pathways and pathways related to protein synthesis/processing and oxidative phosphorylation as being the most significantly downregulated pathways in IPF (Figure 5C). To determine the phenotype of the Lyso^pos^ IPF population, we superimposed the transcriptomic signatures of several relevant cell types from two recent publications onto our differentially expressed gene expression data [29,31]. First, we defined the signatures of all relevant cell types in each data set using the first 30 most differentially expressed genes for each cell type: AEC2, signaling AEC2, basal, differentiating and proliferating basal, AEC1, ciliated and club cells (Travaglini et al.), and AEC2, AEC1, basal, aberrant basal, ciliated and club (Adams et al.). These signatures were then superimposed onto our Donor/IPF LysoTracker comparison, showing an overall downregulation of the AEC2 signature in the IPF Lyso^pos^ population. However, a closer look at the surfactant compartment genes revealed the significant downregulation of several surfactant synthesis and processing genes (*NAPSA*, *ABCA3*, *LAMP3*, *LPCAT1*), while the surfactant protein genes *SFTPB* and *SFTPC* were not significantly regulated (*SFTPC* LFC = −0.25, LOG(p) = 0.24 and *SFTPB* LFC = −0.30, LOG(p) = 0.50). This suggested that the two populations, which homogeneously express proSP-B and proSP-C mRNA (Figure 5D) and protein (Figure 3), differ in the expression of the processing machinery that would normally commit them to a bona fide AEC2 fate. In addition, two fundamental regulators of AEC2 fate had opposite patterns of expression: IPF Lyso^pos^ cells expressed markedly decreased levels of *ETV5* (LFC =−3.42, LOG(p) = 2.32), but *SOX9* (LFC = 3.25, LOG(p) = 8.93) was one of the top overexpressed genes in our data set (Figure 5D and Appendix A). Further analysis showed the upregulation of basal, differentiating basal and aberrant basal transcriptomic signatures in IPF, suggesting the presence of cells belonging to the basal cell lineage. Ciliated, club and AEC1 signatures did not unequivocally superimpose with any of the up or downregulated transcriptomic profiles (Figure 5E and Appendix A). Taken together, this data demonstrates that the Lyso^pos^ population in IPF most likely represents a heterogeneous population of basal-derived cells with the common property of surfactant protein B and C expression but lacking a mature surfactant processing compartment necessary to compensate for the surfactant defects known to occur in IPF.

### 3.5. Basal Cell Marker Expression in Donor and IPF Lung

Our data, in consensus with the existing literature, suggested an increase in basal and aberrant basaloid cells in the distal lung epithelium of IPF patients, which have, in the past, been identified by their intracellular expression of cytokeratin 5 (CK5, the protein product of *KRT5* mRNA) or cell surface expression of NGFR [47,48]. First, we determined by flow cytometry the expression of the CK5 protein in the epithelial compartment of six donor and six IPF lungs. Of note, CK5 expression as a function of HTII-280 demonstrated that HTII-280^pos^ cells did not express CK5 in donor or IPF lung, and the CK5 upregulation was strictly limited to the HTII-280^neg^ compartment (Figure 6A and Appendix A). Indeed, the number of CK5^pos^ cells was greatly increased in the epithelial compartment of IPF lung (average Q3 = 37.86%) compared to donor lung (average Q3 = 12.15% Figure 6A,B and Appendix A). We also determined, in a similar manner, the cell surface expression of NGFR in six donor and six IPF lung cell preparations (Figure 6C,D and Appendix A). While the number of NGFR^pos^ cells also increased significantly in proportion in IPF compared to donor epithelial cells (average 20.1% IPF vs. 3% donor), their proportion was much lower than that of CK5^pos^ cells in both groups (donor and IPF), suggesting the existence of a population of CK5^pos^ cells that do not express NGFR. Of note, in our transcriptomic analysis, NGFR showed levels of expression in both AEC2 populations below the threshold above which a gene was considered to be expressed (Figure 6B and Appendix A). We next asked if the expression of the two markers defines distinctly localized populations of epithelial cells in either donor and/or IPF lung. Thus, we analyzed by immunofluorescence staining the pattern of expression of CK5 and NGFR together with the AEC2 marker ABCA3 in six donor and six IPF lung samples. Representative confocal images (Figure 6E) show that, in both donor and IPF lungs, ABCA3-expressing AEC2s expressed neither CK5 nor NGFR, confirming the flow cytometry data in Figure 6A,C. Additionally, in donor’s lung, extensive areas of basal cells were labeled with either CK5 alone or co-expressed with NGFR (CK5^pos^ NGFR^pos^ cells) in a clonal fashion. In the IPF lung, CK5^pos^ NGFR^pos^ cells were found in either normal-appearing basal cells in the conducting airways or in the simple or pseudo-stratified epithelium lining epithelial cysts in distal fibrotic areas. CK5^pos^ NGFR^neg^ cells were present, predominantly as highly metaplastic areas in the fibrotic distal lung (Figure 6D). Together, these data demonstrate that NGFR expression can differentiate two populations of CK5^pos^ basal cells with different behavior in IPF.

### 3.6. CD24 Upregulation in IPF

CD24 was identified as a cell surface marker highly expressed by aberrant basaloid cells [31] and in the KRT5-/KRT17+ intermediate cells [32] (Figure 7A and Appendix A). In our transcriptomic analysis, CD24 was also markedly increased in IPF Lyso^low^ cells (LFC =2.92, LOG(p) = 3.1). To determine which epithelial cells express CD24, we stained four donor and three IPF lung cell preparations with the usual cell surface markers in combination with a CD24 antibody, and its expression was analyzed in the epithelial compartment as a function of LysoTracker uptake. In donor epithelial cells, the proportion of CD24^pos^ cells was very small (average 1.4% in Lyso^high^ and 3.41% in Lyso^low^ population), but its expression was markedly increased in both IPF Lyso^high^ and Lyso^low^ populations (average 15.2% and 28.5% respectively, Figure 7B–D; individual patient data in Appendix A). Given the basal/AEC2 profile of the Lyso^low^ CD24^pos^ population suggested by our transcriptomic data, our cell surface expression analysis allows us to speculate that the CD24^pos^ Lyso^low^ cells might represent a sub-population of an IPF-specific intermediate cell type.

## 4. Discussion

Here we provide an in-depth phenotypical analysis of human alveolar epithelial cells isolated from donor and end-stage IPF explanted lungs. In doing so, we identify two populations that differ markedly in their ability to process and secrete surfactant, the defining differentiated function of AEC2s. During bleomycin-induced lung fibrosis, the two populations vary in complementary directions, suggestive of correlative behavior. Comparing the transcriptome of the IPF and donor Lyso^pos^ populations in human lung, we determine that the IPF Lyso^pos^ population co-expresses markers of basal and AEC2 lineages. We further confirm by flow cytometry and immunofluorescence analysis the CK5^pos^ cell expansion in IPF and show that CK5 and NGFR expression define two distinct basal cell populations with differential behavior in IPF.

CD24 is a widely expressed glycophospholipid (GPI)-anchored cell surface protein localized to lipid rafts with a versatile signaling ability through cis- and trans-association with various transmembrane receptors [49]. Its epithelial expression was recently identified as the core of the ligand-receptor interactome in the development of human lung adenocarcinoma [50]. Interestingly, in ovarian and breast cancer, CD24 functions as a checkpoint inhibitory molecule, mediating macrophage-phagocytosis evasion through its interaction with Siglec 10 [51]. In IPF, its expression is specifically increased in aberrant epithelial cells, ionocytes, and pulmonary neuroendocrine cells [30,31,32]. Our data confirm the increased cell-surface expression of CD24 in the intermediate Lyso^low^ population in IPF, thus offering a potential cell surface marker to sub-type, together with LysoTracker incorporation, various populations of epithelial cells in donor and IPF lung. A possible correlation between CD24 expression and the well-documented increase in lung adenocarcinoma development in IPF patients is intriguing and remains to be addressed experimentally [50].

The emergence of single-cell transcriptomics led to the in-depth profiling of already known populations of cells and the identification of other novel populations in the mouse and human lung [27,29,30,31,32,52]. Recent landmark papers led to the identification of epithelial populations with an intermediate transcriptomic signature in the IPF lung. First, a population of cells with an “intermediate phenotype” that resembled the transcriptomic profiles of both AEC2s and basal cells was identified by Xu et al. [28]. Recently, a similar population of cells specific for IPF called aberrant basal cells was described by Reyfman et al. and by Adams et al. [30,31].

Similarly, we find that the Lyso^pos^ population in IPF expresses several markers defining these intermediate populations. Characteristic markers, such as KRT5, 15 and 17, ITGA2, Sox4, Sox9, and CD24, are expressed at high levels, together with the AEC2 genes SFTPB and C. Genes involved in surfactant protein processing and secretion are also expressed in the IPF Lyso^pos^ population, although they do not reach the level of expression of the donor AEC2s, which correlates well with the low and intermediate levels of LysoTracker incorporation in this population. Interestingly, two genes defining the AEC2 cell fate, *ETV5* and *SOX9*, are also expressed in the IPF Lyso^pos^ population, but while *SOX9* is expressed at levels exceeding that of bona fide AEC2s, *ETV5* expression is much lower, suggesting that they regulate different aspects of the IPF alveolar epithelial fate. Indeed, *SOX9* is crucial for mouse and human distal lung epithelium specification [53,54,55,56], while *ETV5*, acting downstream of *SOX9* [56,57,58,59] and FGF signaling [60,61], is crucial for AEC2 fate maintenance and is downregulated in transitional states, such as the AEC2 to AEC1 transition [61]. Interestingly, a recent paper identified a population with similar characteristics in the mouse lung [62]. Lineage tracing of alveolar epithelial cells using the SFTPCcreER/Rosa26^TdTomato^ double transgenic mice revealed a TdTomato^low^ population of cells which express low levels of the AEC2 markers proSP-C, *etv5* and *Fgfr2b* that has progenitor cell properties. This suggests that a similar population with intermediate AEC2 characteristics and progenitor properties exists in the mouse lung, which is supported by our data showing the maximum expansion of the Lyso^low^ population at the peak of epithelial proliferation (day 14) following bleomycin injury.

There are multiple circumstances that require the transition through an intermediate fate. First, it has been shown that multiple progenitors can participate in the repair of the alveolar epithelium [63]. It is, thus, conceivable that they converge on a common intermediate state on their way to becoming fully differentiated AEC2s. Second, aberrant progenitors, that in a normal state do not participate in alveolar repair, can be recruited when the local progenitor pool is exhausted, as is the case in the IPF lung, converging towards the same intermediate fate. Third, AEC2 divergent intermediate fates could also emerge, such as (1) AEC2s differentiating into AEC1s [43], (2) AEC2s de-differentiating in order to assume a progenitor function [22] or (3) AEC2s that temporarily limit their differentiated function to allow recovery from injury. Thus, although the Lyso^low^ population appears homogeneous from a phenotypic perspective, as seen by the surfactant protein expression and LysoTracker incorporation, one cannot exclude that it might represent a lineage-diverse population. Based on our data, we propose that the Lyso^low^ population represents a stable cellular state on the way to or from a mature AEC2, rather than a particular cell type. Our population level transcriptomic analysis does not allow us to draw conclusions about the lineage composition or transcriptomic heterogeneity of the Lyso^low^ population, but flow cytometry analysis offers a modality of isolating cells in this intermediate state for further analysis.

In conclusion, we show that LysoTracker incorporation defines two cellular states in donor and IPF distal epithelial lung, with the Lyso^high^ state representing bona fide AEC2s and the Lyso^low^ state characterizing an intermediate cell population displaying both basal and AEC2 characteristics.

## Figures and Tables

**Figure 1 cells-11-00235-f001:**
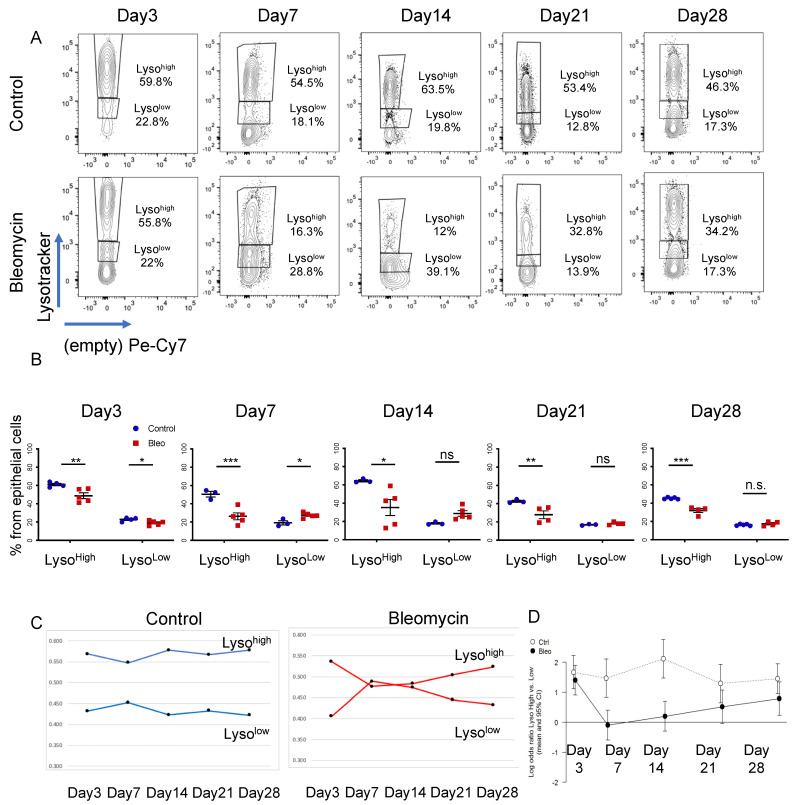
Characterization of AEC2s in a bleomycin model of lung fibrosis. (**A**–**D**) Bleomycin or saline were intratracheally instilled into the lungs of C57B6 mice, which were analyzed after 3 (control *n* = 4, bleomycin *n* = 5), 7 (control *n* = 3, bleomycin *n* = 5), 14 (control *n* = 3, bleomycin *n* = 5), 21 (control *n* = 3, bleomycin *n* = 4) and 28 (control *n* = 4, bleomycin *n* = 4) days. The flow cytometry analysis of the AEC2s in the epithelial compartment, defined as DAPI^−^ CD45^−^ CD31^−^ PDP^−^ EpCAM^+^. (**A**) Representative panels of LysoTracker uptake (Lyso^high^ and Lyso^low^) as a percentage of the parent epithelial compartment of bleomycin-treated mice. (**B**) Statistical analysis of the Lyso^high^ and Lyso^low^ populations at each time point. Data are presented as the means ± SEM. * *p* < 0.05, ** *p* < 0.01, *** *p* < 0.001, n.s. = not significant by ANOVA. (**C**) Time-course analysis of the Lyso^high^ and Lyso^low^ populations in control mice (left panel) and bleomycin-treated mice (right panel). (**D**) Analysis of the log odds ratio of the Lyso^high^ vs. Lyso^low^ population (log Lyso^high^/Lyso^low^) during the time course of bleomycin recovery. Data are presented as the means and 95%confidence intervals.

**Figure 2 cells-11-00235-f002:**
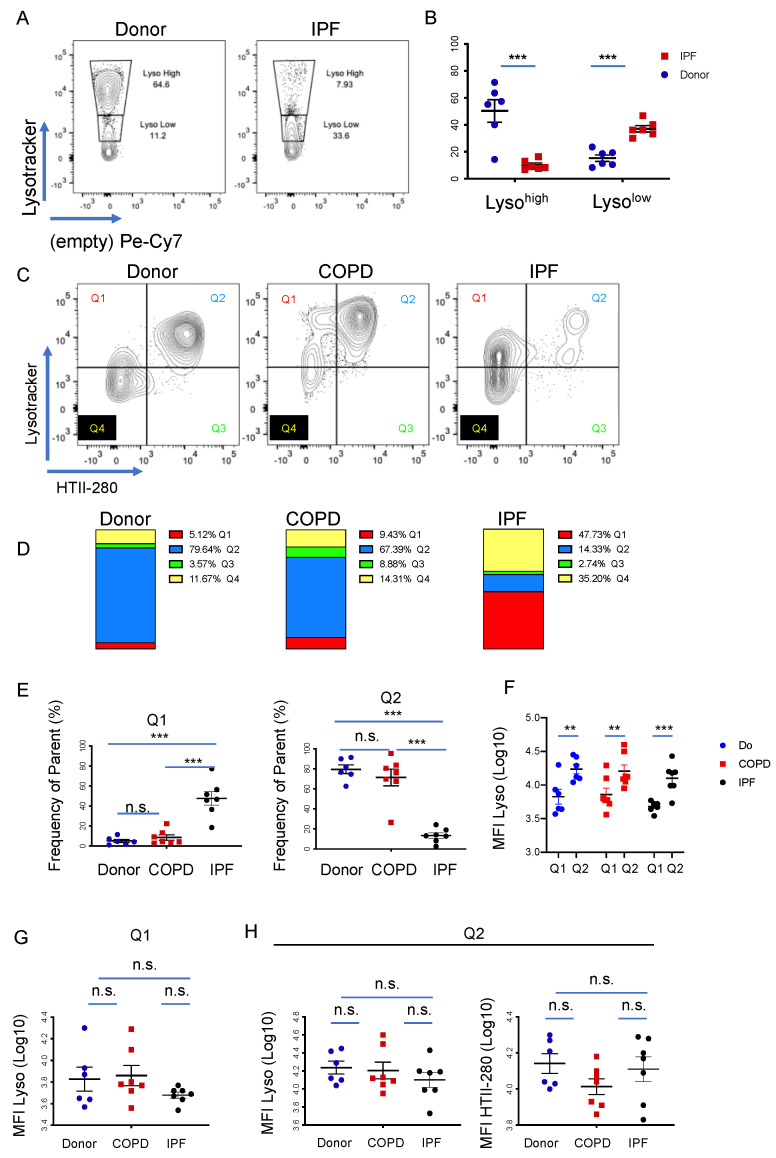
LysoTracker uptake in the epithelial compartment of the human lung. (**A**) Representative panels of flow cytometry analysis of the LysoTracker uptake (Lyso^high^ and Lyso^low^) in the epithelial compartment (DAPI^−^ CD45^−^ CD31^−^ EpCAM^+^) of human donor (*n* = 6) and IPF (*n* = 6) lungs. (**B**) Quantification of the Lyso^high^ and Lyso^low^ populations in donor and IPF samples in (**A**). (**C**) Representative panels of LysoTracker uptake (*y*-axis) as a function of HTII-280 reactivity (*x*-axis) in the epithelial compartment of donor (*n* = 6), COPD (*n* = 7) and IPF (*n* = 7) lungs. Quadrant gating identifies four different populations as follows: Q1 (Lyso^pos^/HTII-280^neg^), Q2 (Lyso^pos^/HTII-280^pos^), Q3 (Lyso^neg^/HTII-280^pos^) and Q4 (Lyso^neg^/HTII-280^neg^). (**D**) Quantification of the data shown in (**C**) showing the relative contribution of the Q1 to Q4 populations to the epithelial compartment of donor, COPD and IPF lungs. (**E**) Quantification of Q1 (left diagram) and Q2 (right diagram) as the frequency of the parent (DAPI^−^ CD45^−^ CD31^−^ EpCAM^+^) population. (**F**) Comparison of the LysoTracker uptake in the LysoTracker-positive populations, Q1 and Q2, in donor, COPD and IPF patients, as measured by the MFI of the respective populations. (**G**) Comparison of the LysoTracker uptake in the Q1 population in donor, COPD and IPF patients, as measured by the MFI of the respective populations. (**H**) Comparison of the LysoTracker uptake (left panel) and HTII280 reactivity (right panel) in the Q2 population in donor, COPD and IPF patients, as measured by the MFI of the respective populations. Data are presented as the means ± SEM of the percentage of cells from the parent population. Statistical analysis was performed on log(10) values. ** *p* < 0.01, *** *p* < 0.001, n.s. = not significant by ANOVA.

**Figure 3 cells-11-00235-f003:**
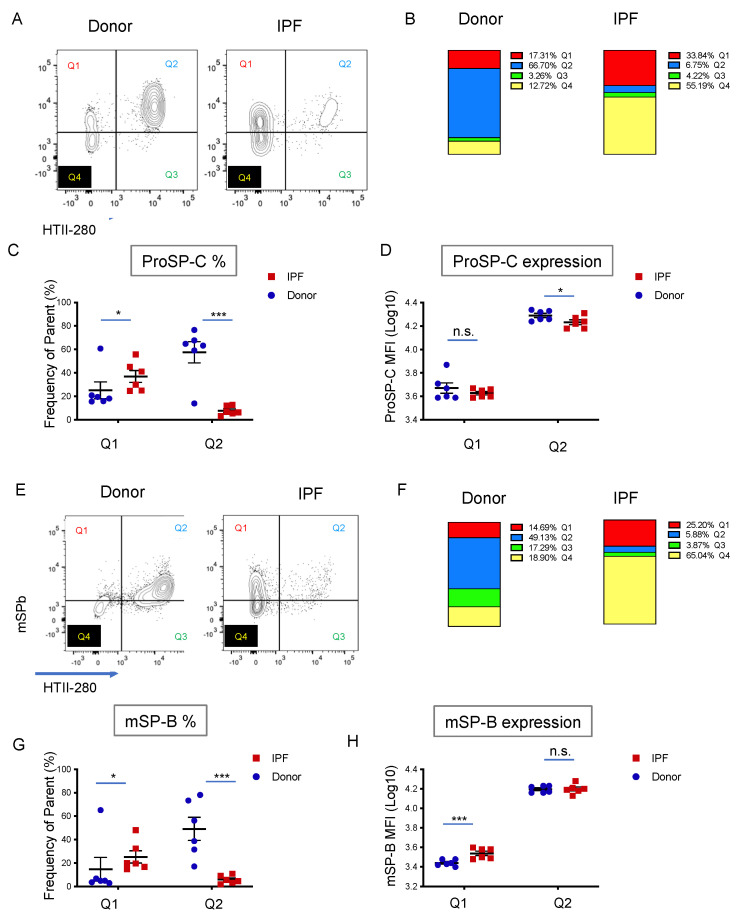
Surfactant protein expression in the epithelial compartment of donor and IPF lung. (**A**) Representative flow cytometry panels of proSP-C and HTII-280 expression in the epithelial compartment (DAPI^−^ CD45^−^ CD31^−^ EpCAM^+^) of donor (*n* = 6) and IPF (*n* = 6) lung preparations. (**B**) Average contribution of the Q1–Q4 populations to the epithelial compartment of the samples shown in (**A**), showing the change in epithelial composition in IPF lung compared to donors. Left column: donors; right column: IPF. (**C**) Quantification of the population frequency of Q1 and Q2 in donor (blue dot) and IPF (red square) lung samples shown in (**A**). (**D**) Quantification of the MFI as a measure of proSP-C expression level (log10 MFI) in the Q1 and Q2 populations of the samples shown in (**A**). (**E**) Representative flow cytometry panels of mSP-B and HTII-280 expression in the epithelial compartment (DAPI^−^ CD45^−^ CD31^−^ EpCAM^+^) of donor (*n* = 6) and IPF (*n* = 6) lung preparations. (**F**) Average contribution of the Q1–Q4 populations to the epithelial compartment of the samples shown in (**E**). Left column: donors; right column: IPF. (**G**) Quantification of the population frequency of Q1 and Q2 in donor and IPF lung samples in (**E**). (**H**) Quantification of the MFI as a measure of mSP-B expression level in the Q2 population of the samples in (**E**). Data are presented as the means ± SEM of the log10 (MFI) values. * *p* < 0.05, *** *p* < 0.001, ns = not significant by Student *t*-test.

**Figure 4 cells-11-00235-f004:**
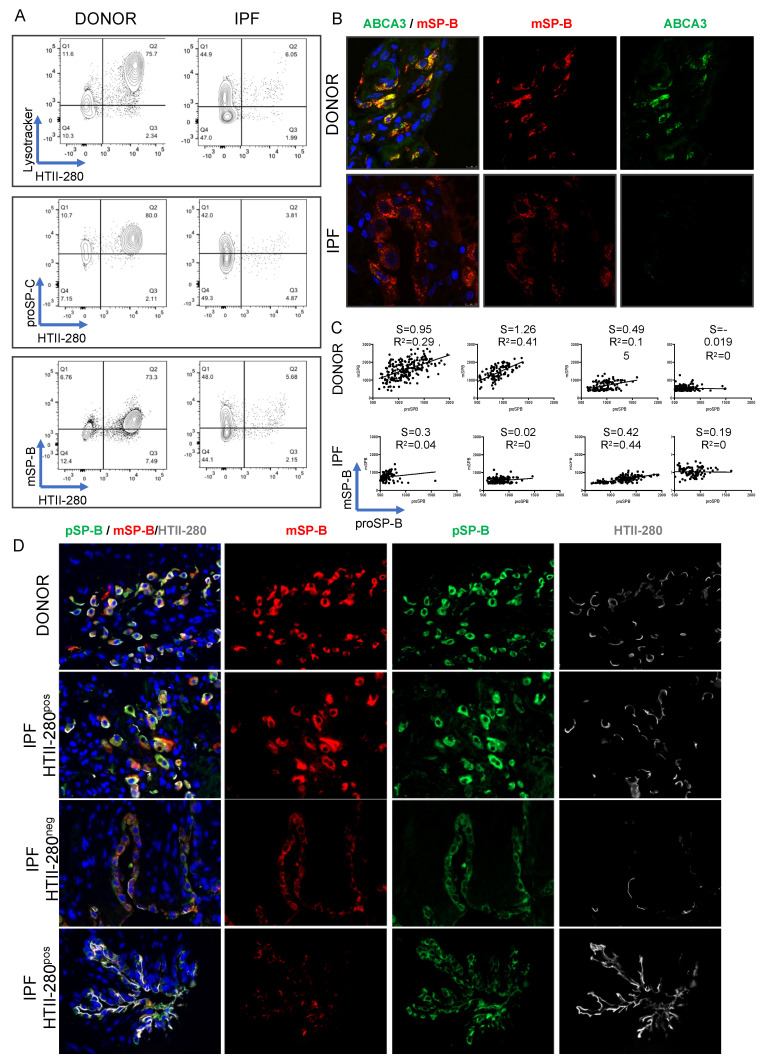
Comparative expression of LysoTracker, proSP-C and mSP-B expression in donor and IPF lung. (**A**) Six donor and six IPF lung preparations were co-stained in parallel with HTII-280, LysoTracker, proSP-C and mSP-B. Representative flow cytometry panels of LysoTracker (upper), proSP-C (middle) and mSP-B (lower) vs. HTII-280 expression in the epithelial compartment of one donor (left column) and one IPF (right column) lung. (**B**) Representative immunofluorescence images of mature (red) and ABCA3 (green) in donor and IPF paraffin-embedded lung tissues. (**C**) Quantification of the fluorescence intensity of the mature SP-B and proSP-B immunofluorescence signals in four donor (upper row) and four IPF (lower row) patients, showing the slope (s) of the linear regression and the correlation index (R^2^) for each patient. (**D**) Representative immunofluorescence images of the mature (red) and proSP-B (green) in donor and IPF tissues shown in (**C**).

**Figure 5 cells-11-00235-f005:**
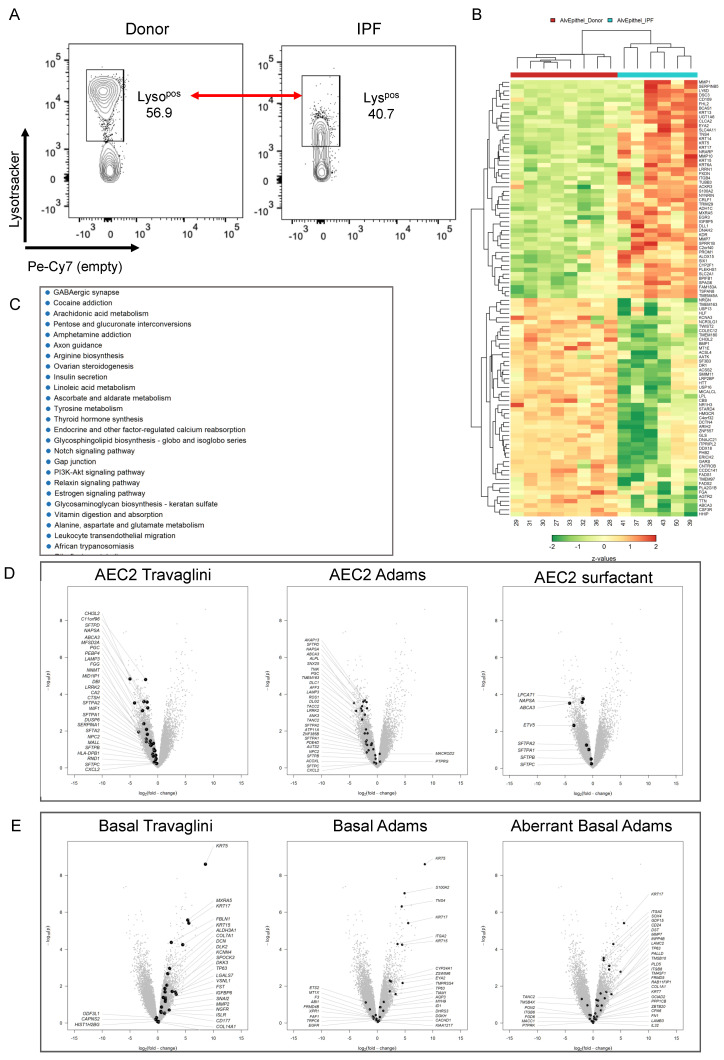
Transcriptomic profiling of the Lyso^pos^ population in IPF. Donor Lyso^pos^ (*n* = 8) and IPF Lyso^pos^ (*n* = 6) epithelial cells were isolated by flow cytometry and their transcriptomic profiles were determined by microarray analysis. (**A**) Flow cytometry panels showing the sorting strategy for the two populations. (**B**) Heat map of the first upregulated and downregulated genes in IPF Lyso^pos^ vs. donor Lyso^pos^. (**C**) KEGG pathway analysis showing the first 20 most differential regulated pathways. (**D**) The transcriptomic signatures of AEC2 that were identified by Travaglini et al. and Adams et al. were superimposed on the vulcano plots depicting the upregulated and downregulated genes in IPF Lyso^pos^ compared to donor Lyso^pos^ (left and middle plots). The right plot shows the distribution of surfactant production and processing genes. (**E**) The transcriptomic signatures of various populations of donor and IPF basal cells that were identified by Travaglini et al. and Adams et al. were superimposed on the vulcano plots depicting the upregulated and downregulated genes in IPF Lyso^pos^ compared to donor Lyso^pos^.

**Figure 6 cells-11-00235-f006:**
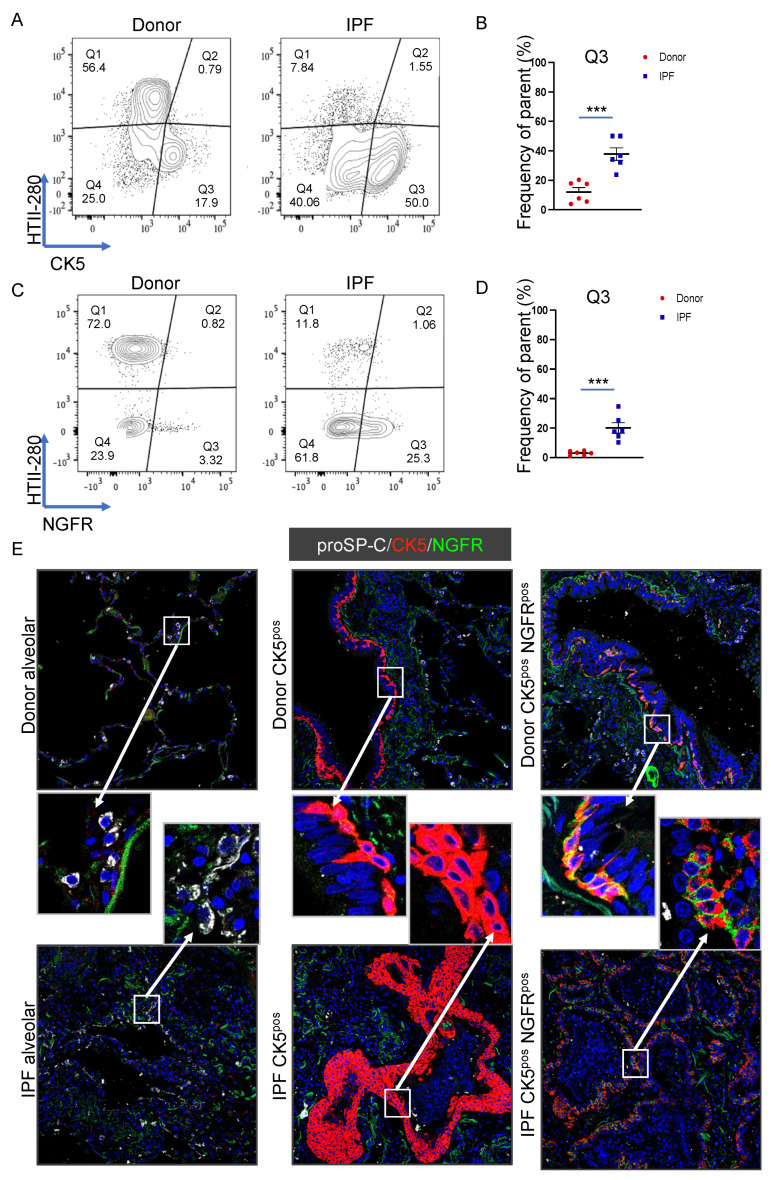
CK5 and NGFR expression in donor and IPF epithelial cells. (**A**) Representative flow cytometry panels of CK5 vs. HTII-280 expression in the epithelial cell compartment of donor (*n* = 6, left) and IPF (*n* = 6, right) lungs. (**B**) Quantification of the CK5^pos^ HTII-280^neg^ (Q3) population shown in (**A**). (**C**) Representative flow cytometry panels of NGFR vs. HTII-280 expression in the epithelial cell compartment of donor (*n* = 6, left) and IPF (*n* = 6, right) lungs. (**D**) Quantification of the NGFR^pos^ HTII-280^neg^ (Q3) population shown in (**C**). Data are presented as the means ± SEM of the percentage of cells from the parent population. Statistical analysis was performed on log(10) values. *** *p* < 0.001, n.s. = not significant by ANOVA. (**E**) Representative confocal images of proSP-C (white signal), CK5 (red signal) and NGFR (green signal) in different locations of donor (upper images) and IPF (lower images) lung.

**Figure 7 cells-11-00235-f007:**
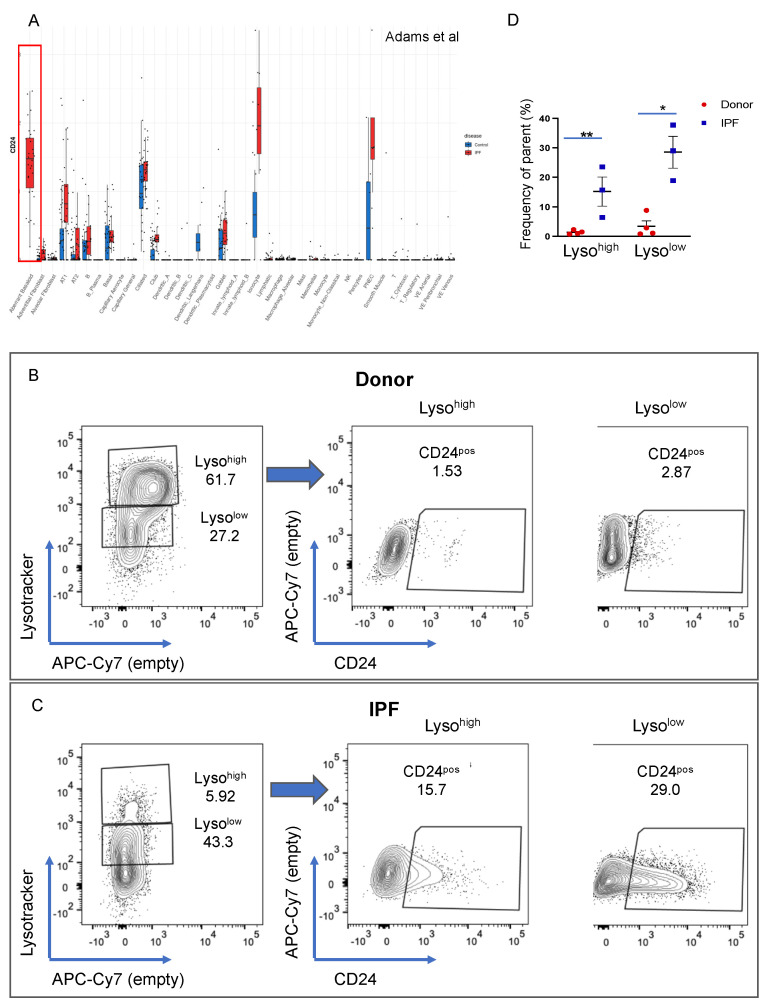
CD24 expression in donor and IPF lung. (**A**) Differential expression of CD24 in donor and IPF scNGS data published by Adams et al. (**B**), (**C**) Flow cytometry analysis of donor (n = 4) (**B**) and IPF *(n* = 3) (**C**) samples of LysoTracker incorporation in the epithelial cell compartment (left panel). Right panels show the expression of CD24 in the Lyso^high^ and Lyso^low^ populations shown in the panels on the left. (**D**) Quantification of the data in (**B**,**C**) showing the difference between donor and IPF Lyso^high^ and Lyso^low^ populations. Average donor Lyso^high^ CD24^pos^ 1.4%, donor Lyso^low^CD24^pos^ 0.34%; IPF Lyso^high^ CD24^pos^ 15.2% and IPF Lyso^low^ CD24^pos^ 28.53%. Data are presented as the means ± SEM of the percentage of cells from the parent population. Statistical analysis was performed on log(10) values. * *p* < 0.05, ** *p* < 0.01, n.s. = not significant by ANOVA.

## Data Availability

Transcriptomic data is publicly available at the following location: https://www.ncbi.nlm.nih.gov/geo/query/acc.cgi?acc=GSE185691.

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
