# Peer review of "Differential LysoTracker Uptake Defines Two Populations of Distal Epithelial Cells in Idiopathic Pulmonary Fibrosis"

_cells, 2022, doi:10.3390/cells11020235_

Round 1
Reviewer 1 Report
The manuscript by Wasnick et al. demonstrate the feasibility of using lysotracker uptake to define two distinct populations of epithelial cells in IPF: the Lyso high population represent the classical AEC2s, and the Lyso low population resembles the previously identified intermediate AEC2 population in the IPF lung. The experiments are sophisticated designed, thorough analyses were performed to characterize the expression of HTII-280, proSP-C, mature SP-B, NGFR, KRT5 and CD24 in lyso high vs lyso low populations. However, a few noted weaknesses are:
- The manuscript starts with mouse bleomycin data to describe the two distinct populations. However, no further characterization was done in mice to verify the findings in human cells. It feels like a disconnection between this part of the manuscript from the rest.
- Lyso tracker flow gating criteria varied a lot in Figure 1. It appears that the authors were basically arbitrarily making the calls.
- Figure 3A, the authors found that there was a statistically significant increase in the proSP-Cneg HTII-280neg (Q4) population which suggested the increased presence of non-AEC2 cells in the distal IPF lung. Similar findings were noted in Fig 3E where Q4 cells were increased in IPF. Can the authors elaborate what these cells were?
- In figure 3E, the authors stated that “mSP-B was expressed in the majority of the bona-fide AEC2s (Q2: HTII-280pos mSP-Bpos) in both donor and IPF”. It does not appear to be true, as in IPF, most mSP-B positive cells are negative for HTII-280.
In fact, as noticed by the authors, the IPF Q1 (HTII-280neg mSP-Bpos) population expressed higher levels of mSPB than donor Q1, suggesting an up-regulation of the surfactant processing ability in this population in disease conditions. These cells are presumably negative for ABCA3. How would they be able to process SPB maturation in a more efficient manner?
- Figure 4C, there was no labeling of donor vs. IPF.
- In Figure 4D, the authors stated “a heterogenous distribution of areas of co-localization (particularly in non-affected areas) and areas where the mSP-B was low or absent in cells that expressed the pro- forms”. Not sure what the authors were referring to in the images. Adding arrows to specifically point out the areas would be helpful.
Author Response
We, the authors, would like to thank the reviewers and the editor for taking the time to review our work and offer their constructive advice. Hoping to bring more clarity to our manuscripts, we responded to the best of our knowledge to each of their questions in the section below. All changes were marked in the manuscript accordingly.
Reviewer 1
The manuscript by Wasnick et al. demonstrate the feasibility of using lysotracker uptake to define two distinct populations of epithelial cells in IPF: the Lyso high population represent the classical AEC2s, and the Lyso low population resembles the previously identified intermediate AEC2 population in the IPF lung. The experiments are sophisticated designed, thorough analyses were performed to characterize the expression of HTII-280, proSP-C, mature SP-B, NGFR, KRT5 and CD24 in lyso high vs lyso low populations. However, a few noted weaknesses are:
- The manuscript starts with mouse bleomycin data to describe the two distinct populations. However, no further characterization was done in mice to verify the findings in human cells. It feels like a disconnection between this part of the manuscript from the rest.
Response 1. The existence of transitional states and their equivalence between mouse and human lung is an area of great interest[1]. The scarcity of human tissue, the inability to use in vivo genetic manipulation tools, and the genetic and phenotypic variability that underlies our species diversity, make mouse models an indispensable and extremely valuable tool in understanding our biology. We considered that our bleomycin data provides enough evidence that the existence of the two populations is causally related to the bleomycin injury and the ensuing repair process. Thus, we provide a useful tool for the research community to further study this subpopulation. We are certain that further single cell transcriptomic and lineage tracing of different populations will bring valuable insight into the lineage composition of the Lysohigh and the Lysolow population, but this is beyond the scope of the current publication.
- Lyso tracker flow gating criteria varied a lot in Figure 1. It appears that the authors were basically arbitrarily making the calls.
Response 2. Figure1 represents the analysis of 5 individual experiments (one per time-point). When working with live cells, mouse lungs have to be dissociated, stained and analyzed by flow cytometry in a timely manner, making it difficult to process a very high number of samples per experiment. Moreover, variability in the cell isolation, staining, lysotracker incorporation procedure and instrument performance, make it very difficult to use the same gating strategy throughout multiple individual experiments. Thus, we designed our experiment so that each time point represents a self-sufficient experiment, with its own gating and staining controls, and each data set was analyzed and gated based on the specific controls belonging to that sample set. Most importantly, within each time-point experiment, the same gating strategy was applied to both control and experimental samples. This explains the differences in gating from time-point to time-point.
- Figure 3A, the authors found that there was a statistically significant increase in the proSP-Cneg HTII-280neg (Q4) population which suggested the increased presence of non-AEC2 cells in the distal IPF lung. Similar findings were noted in Fig 3E where Q4 cells were increased in IPF. Can the authors elaborate what these cells were?
Response 3. Although chronic AEC injury is the underlying cause of IPF, all other cell types in the lung participate in a reactive manner to the disease pathogenesis. It is well known that conducting epithelial cells like basal, goblet, club and ciliated cells are all increased in number in IPF. This phenomenon was well described at single cells and tissue level, using transcriptomic or histological assessment[2–5].
- In figure 3E, the authors stated that “mSP-B was expressed in the majority of the bona-fide AEC2s (Q2: HTII-280pos mSP-Bpos) in both donor and IPF”. It does not appear to be true, as in IPF, most mSP-B positive cells are negative for HTII-280.
In fact, as noticed by the authors, the IPF Q1 (HTII-280neg mSP-Bpos) population expressed higher levels of mSPB than donor Q1, suggesting an up-regulation of the surfactant processing ability in this population in disease conditions. These cells are presumably negative for ABCA3. How would they be able to process SPB maturation in a more efficient manner?
Response 4. (1) The statement “mSP-B was expressed in the majority of the bona-fide AEC2s (Q2: HTII-280pos mSP-Bpos) in both donor and IPF” refers to the level of mature SP-B in the HTII-280pos population. Although in IPF most mSP-Bpos cells are indeed negative for HTII-280, our statement refers to the mSP-B expression in the HTII-280pos population. To clarify this statement, we included a specific analysis of mSP-B expression in the HTII-280pos population in donor and IPF epithelium as Supplemental Figure 3D in the revised manuscript.
(2) Although we have not studied in detail the mechanism of surfactant processing upregulation, we can speculate that the increase expression of mSP-B in the IPF Q1 is the result of upregulation of alternative proteases. In our microarray comparison of Lysotracker positive cells, IPF AEC2s express increased levels of cathepsin E, O and S.
- Figure 4C, there was no labeling of donor vs. IPF.
Response 5. Corrected, thank you!
- In Figure 4D, the authors stated “a heterogenous distribution of areas of co-localization (particularly in non-affected areas) and areas where the mSP-B was low or absent in cells that expressed the pro- forms”. Not sure what the authors were referring to in the images. Adding arrows to specifically point out the areas would be helpful.
Response 6. Thank you for pointing out this unclear statement. We are actually referring to the different rows of panels in Figure 4D. To clarify, we re-arranged the panels in Figure 4 D, where we moved the row of IPF HTII-280pos panels (normal appearing areas) on the second position. In text we referred to this figure as follows:
“Analyzing the spatial distribution of the two signals, we noticed that in donors they were tightly co-expressed (Figure 4D upper panels), but in IPF there was a heterogenous distribution of areas of mature and proSP-B co-localization (particularly in non-affected areas – Figure 4D panels in rows 2 and 4) and areas where the mSP-B was low or absent in cells that expressed the pro- forms (Figure 4D – panels in row 3).”
Reviewer 2 Report
The authors carefully analyzed and investigated the alveolar epithelial type 2 cells through Lysotracker incorporation by flow cytometry and immunofluorescence approach able to distinguish two populations of distal epithelial cells in Idiopathic Pulmonary Fibrosis. The manuscript was well written, and the topic could be very interesting to improve the knowledge on IPF pathogenesis. Here my comments/suggestions:
- please report the acronym IPF as "Idiopathic Pulmonary Fibrosis" instead of "idiopathic lung fibrosis"
- I suggest the authors to carefully state the aim of the present study at the end of introduction section
- the methods and statistical analysis were correctly performed
- The authors identified increased CD24 expression in IPF. CD24 is expressed at the surface of most B lymphocytes, could authors discuss this result taking into account the potential relationship between IPF pathogenesis and the immunity system?
Author Response
We, the authors, would like to thank the reviewers and the editor for taking the time to review our work and offer their constructive advice. Hoping to bring more clarity to our manuscripts, we responded to the best of our knowledge to each of their questions in the section below. All changes were marked in the manuscript accordingly.
The authors carefully analyzed and investigated the alveolar epithelial type 2 cells through Lysotracker incorporation by flow cytometry and immunofluorescence approach able to distinguish two populations of distal epithelial cells in Idiopathic Pulmonary Fibrosis. The manuscript was well written, and the topic could be very interesting to improve the knowledge on IPF pathogenesis. Here my comments/suggestions:
- please report the acronym IPF as "Idiopathic Pulmonary Fibrosis" instead of "idiopathic lung fibrosis".
Thank you for pointing this out, we changed the text accordingly.
- I suggest the authors to carefully state the aim of the present study at the end of the introduction section.
The last paragraph of the introduction was modified to include the reviewer’s suggestions as follows:
“Here, we ask if the aberrant or intermediate transcriptional programs recently identified in IPF[2,3,6] result in functional transitional states that can be identified by the low/intermediate ability to process and secrete surfactant proteins (SP). To that end, we analyze the size of the surfactant processing compartment in dissociated human donor and IPF lung epithelial cells, thus defining two functional alveolar epithelial states present in both donor and IPF lung. Based on known intra-cellular, cell surface proteins and lysotracker incorporation, coupled with transcriptomic analysis, we show that the Lysotrackerhigh population consists of bonafide AEC2s and is drastically diminished in IPF. A second population of Lysotrackerlow cells, which uniformly expresses and processes surfactant proteins, but bears the transcriptional footprint of a CK5-derived (basal) population is increased in IPF.”
- the methods and statistical analysis were correctly performed
- The authors identified increased CD24 expression in IPF. CD24 is expressed at the surface of most B lymphocytes, could authors discuss this result taking into account the potential relationship between IPF pathogenesis and the immunity system?
In response to the reviewer’s request, we added the following paragraph in the Discussions section of the manuscript:
“CD24 is a widely-expressed glycophospholipid (GPI) anchored cell surface protein localized to lipid rafts with versatile signaling ability through cis- and trans- association with various transmembrane receptors[51]. Its epithelial expression was recently identified as the core of the ligand-receptor interactome in the development of human lung adenocarcinoma [52]. Interestingly, in ovarian and breast cancer, CD24 functions as a check-point inhibitory molecule, mediating macrophage-phagocytosis evasion through its interaction with Siglec 10 [53]. In IPF, its expression is specifically increased in aberrant epithelial cells, ionocytes, and pulmonary neuroendocrine cells [31,34]. Our data confirm the increased cell-surface expression of CD24 in the intermediate Lysolow population in IPF, thus offering a potential cell surface marker to sub-type together with Lysotracker incorporation various populations of epithelial cells in donor and IPF lung. A possible correlation between CD24 expression and the well-documented increase in lung adenocarcinoma development in IPF patients is intriguing and remains to be addressed experimentally.”